# Ovarian Dynamics and Changes in Estradiol-17β and Progesterone Relationship with Standing Estrus, Preovulatory Follicles, and Ovulation Using Single Prostaglandin F2α Injection in Barbari Goats

**DOI:** 10.3390/vetsci10100624

**Published:** 2023-10-19

**Authors:** Tariq Sohail, Muhammad Farhab, Liuming Zhang, Yan Kang, Xiaomei Sun, Dejun Ji, Yongjun Li

**Affiliations:** 1Key Laboratory for Animal Genetics & Molecular Breeding of Jiangsu Province, College of Animal Science and Technology, Yangzhou University, Yangzhou 225009, China; drtariqsohail34@yahoo.com (T.S.);; 2Department of Theriogenology, University of Veterinary and Animal Sciences, Lahore 54000, Pakistan; 3Key Laboratory of Animal Genetic Engineering, College of Veterinary Medicine, Yangzhou University, Yangzhou 225009, China; farhab.dvm@gmail.com

**Keywords:** Barbari goats, PGF2α, ovarian dynamics, hormonal profile, estrus, ovulation

## Abstract

**Simple Summary:**

There is an enormous goat population in Asia, especially in China, India, and Pakistan. Different estrus synchronization protocols such as Ovsynch and PGF_2α_ are widely used in reproductive biotechnology to optimize estrus and ovulation in small ruminants to improve reproductive efficiency. Efficient small ruminant production depends upon the knowledge of re productive physiology. This study provides knowledge about the correlation among physical and behavioral signs of estrus with ovarian dynamics and hormonal profiles such as estradiol-17β and progesterone after induced luteolysis during the preovulatory period in Barbari goats. The findings of this study will be helpful in developing better estrus synchronization strategies for improving the reproductive efficiency of Barbari goats.

**Abstract:**

The purpose of the present research was to define ovarian follicular dynamics and plasma endocrine profiles in response to a single PGF_2α_ injection, administered indiscriminately during the breeding season of Barbari goats. Ovarian dynamics were observed at every 12 h interval by using B mode ultrasonography, blood samples for hormonal analysis such as estradiol 17β and progesterone were collected at every 12 h interval, and bucks with aprons were used to identify standing estrus at every 6 h interval. Relative to PGF_2α_, the start of standing estrus and ovulation differ (*p* < 0.05) between early- (n = 7), intermediate- (n = 6), and late-responding (n = 6) goats. The highest plasma level of estradiol 17β was detected 12 h prior to ovulation. The average diameter of the ovulatory follicle and length of standing estrus were comparable (*p* > 0.05) between the goats. The corpus luteum degenerated more quickly (*p* < 0.05) in early- than intermediate- and late-responding goats. Dominant follicle diameter and estradiol 17β concentration also differ (*p* < 0.05) among groups. Although the plasma level of progesterone did not vary (*p* = 0.065), the variation in progesterone concentration with time differed (*p* < 0.05) amongst the goats. As a result, this research indirectly reveals that the beginning of standing estrus, end of estrus, and ovulation after PGF_2α_ might fluctuate in Barbari goats because of follicular and hormonal dynamics during the luteal phase.

## 1. Introduction

There is continued demand for increased meat production in the Asian subcontinent due to the enormous human population. In this circumstance, an efficient animal production system is required to ensure food security. Approximately 46.9% of the total world goat population belongs to China, India, and Pakistan (FAO, 2017) [1]. Barbari goats are a major component of livestock rearing in SAARC countries, especially Pakistan, India, Sri Lanka, and Afghanistan [2]. The Barbari goat is a medium-sized dual-purpose (milk and meat) breed renowned for good feed conversion efficiency, high reproductive efficiency, high prolificacy, early maturity, high kidding rate, and natural resistance to various bacterial and parasitic diseases [3,4,5]. The Barbari goat is also highly adaptive to harsh climatic conditions in arid and semi-arid regions in South Asian countries due to the expression of heat shock protein genes HSP60 and HSP70 [6].

In small ruminants, prostaglandin F_2α_ (PGF_2α_) and its analogs are extensively used for estrus synchronization during the breeding season [7]. Prostaglandins have the benefit of being easy to inject intramuscularly, distribute rapidly, and almost completely metabolize in the lungs [8]. Estrus can be induced in >70% vs. >90% of goats using either single or double injections (11–14 days apart) of PGF_2α_ during the breeding season of goats [9]. The efficiency of PGF_2α_ depends on time, dose, route of administration, season, male effect, and ovarian follicular dynamics; however, the presence of the corpus luteum is indispensable for successful estrus induction [10,11,12,13].

In ewes and goats, PGF_2α_ does not induce luteolysis when administered prior to day 4 or on day 17 of the estrous cycle [14,15]. During the luteal phase, a variable estrous response (40–80%) to a single PGF_2α_ injection has been observed in ewes, primarily due to the variable developmental stage of the corpus luteum [16]. Similarly, in dairy goats, estrous response ranged from 67 to 85% when PGF_2α_ was injected on day 6 or 12 of the estrous cycle [7]. The time of ovulation in Beetal goats fluctuates between 60 and 96 h, with the majority of ovulation occurring approximately 72 h following a single PGF_2α_ treatment. Hence, the peak plasma concentration of estradiol 17β and luteinizing hormone (LH) and the decline in progesterone concentration, standing estrus, dominant follicles, and corpus luteum diameter also significantly vary in early- and late-ovulating Beetal goats [17].

In another study, an early start of estrus was detected when goats were administered with PGF_2α_ at day 5 and then on days 11 and 16 of the estrous cycle. Altogether, findings from the literature indicate that both estrous responses as well as the interval to the onset of estrus following PGF2α vary due to the time it takes for the synchronization protocols to begin during the luteal phase [18]. Fluctuations in the peripheral plasma concentration of estradiol 17β, progesterone, and luteinizing hormone during the follicular phase have been correlated with the onset of estrus and ovulation in goats, ewes, lactating cows, and buffalo [19,20,21,22].

The Barbari goat is a spontaneous ovulator and is highly fertile, with twining and tripling birth and an ovulation rate of 1.43; extrinsic factors such as feed and environmental factors also affect reproductive efficiency [23,24]. The effects of a dietary intake of rice dried distilled grain (rDDGS) on the physical and behavioral signs of estrus in Barbari goats were observed [25]. Some studies also highlight genetic and non-genetic factors affecting growth, productivity, and reproductive characteristics of Barbari goats raised in semi-intensive care systems [2,4]. Previous studies also provide information about the relationship between caprine pregnancy-associated glycoprotein (caPAG) and gestation, fetal number, and parity in Barbari goats [26,27]. However, the above-mentioned studies do not provide any reliable information about follicular dynamics and variations in the plasma concentration of estradiol 17β and progesterone in relation to the onset of estrus or PGF_2α_. Therefore, the aim of the current research was to define the onset of standing estrus, interval to ovulation, and changes in preovulatory follicles in relation to the plasma concentrations of estradiol 17β and progesterone following PGF_2α_ treatment in Barbari goats.

## 2. Materials and Methods

### 2.1. Animals and Management

This experiment was conducted at the Small Ruminant Training and Research Centre (SRT&RC), Pattoki, University of Veterinary and Animal Science, Lahore (31°03′29.0″ N 73°52′42.9″ E). The entire procedure performed during the experiment was in accordance with the Animal Care and Ethical Review Committee of the University of Veterinary and Animal Science (UVAS), Lahore, Pakistan, protocol no. 6853, and Yangzhou University, SYXK [Su] 2017-0044. Overall, nineteen healthy, multiparous, cycling goats with body condition score within the range 2.5 to 4 and body weight within the range 30 to 40 kg were raised in an unrestricted stall and were given green fodder (sorghum: 4–5 kg/animal/day), concentrate with corn gluten, corn grain, wheat bran, soybean meal, canola meal (400–500 gm/animal/day), and water ad libitum. This experiment was conducted during the breeding season (September–December 2021).

### 2.2. Selection and Treatment of Animals

These nineteen goats were selected from a flock of thirty goats based on detection of the corpus luteum via ultrasonography. All goats (n = 19) were administered a single dose of PGF_2α_ (Delmazine^®^, Fatro, Italy; 1 mL, i.m) randomly upon detection of the corpus luteum. Ovaries were examined transrectally every 12 h interval after PGF_2α_ (0 h) up to ovulation using a 7.5 MHz transducer (Honda^®^ 1600, Tokyo, Japan) as defined earlier [28]. Ovulation was determined by the rapid vanishing of earlier identified dominant follicles on a consecutive examination 12 h later [29]. Standup estrus was noticed using bucks with aprons after every 6 h. A goat was considered in standing estrus when it stood unmoving and permitted the buck to mount at a given time point [30].

### 2.3. Experimental Design and Methodology

This experiment was conducted during breeding season from September to December 2021. Overall, fifty (n = 50) Barbari goats at research and training station Pattoki were scanned to select appropriate goats for further experimentation. From (n = 50) goats, twenty goats (n = 20) were already pregnant and inappropriate for the experiment. Eleven goats (n = 11) were in the very early (≤4 days) or late (≥17 days) luteal phase, and some were already in estrus and were excluded from consideration. Nineteen goats (n = 19) used in experiments were physically healthy, free from any history of reproductive disorder, and multiparous with an age of approximately 3.5 ± 0.3 years. A single shot of PGF_2α_ was administered upon detection of the mature corpus luteum between (4–16) days, and the 1st blood sample was collected for hormonal analysis. The second blood sample was collected after a 24 h interval, and subsequent sampling was conducted every 12 h interval until ovulation. Ultrasound was performed every 12 h interval to define ovarian follicular and luteal dynamics throughout preovulatory time until ovulation. Bucks with aprons were used to distinguish goats in standing estrus every 6 h interval. These goats (n = 19) were further divided into early- (n = 7), intermediate- (n = 6), and late-ovulating (n = 6) goats due to the significant variation in onset of estrus and ovulation because of different ovarian and hormonal status at the time of synchronization protocol.

In general, research related to reproductive physiology and biotechnology is conducted with a large herd size. We selected n = 19 goats, which could be considered a reasonable sample size if not for the fact that goats respond significantly differently from each other due to variable ovarian status. In addition, the small sample size of this research was due to funding limitations and management and seasonality issues in Barbari goats. The scope of the research could definitely be increased if it is conducted with a larger herd size with short blood sampling intervals for hormonal analysis and ultrasonography for more precise information about hormonal profile association with ovarian dynamics, estrus, and ovulation.

### 2.4. Blood Sampling for Hormonal Analyses

Blood samples for the endocrine profile test were collected at 0, 24, 36, 48, 60, 72, 84, and 96 h following PGF_2α_ administration from the jugular vein in 5 mL vacutainer tubes containing EDTA. Immediately after collection, all samples were centrifuged at 3000× *g* for 15 min to isolate plasma, which was kept at −20 °C until further use. A double antibody radioimmunoassay method was used to estimate the plasma concentration of estradiol 17β and progesterone (Immuno Tech^®^, Beckman Coulter, Prague, Czech Republic) as defined earlier [31]. These kits were standardized for goats. The intraassay coefficients of variation for progesterone and estradiol 17β were 7.6 and 13.8%, respectively. The interassay measurements of deviation for progesterone and estradiol 17β were 8.4 and 14.4%. The lowest detection for the progesterone and estradiol 17β tests remained 0.03 ng/mL and 0.02 pg/mL, respectively.

### 2.5. Statistical Analyses

The data are presented as means ± SEM, and statistical significance was measured at *p* ≤ 0.05. Data were normalized before comparing the means. Data of the corpus luteum and progesterone were normalized with 1/x transformation, while estradiol 17β data were normalized by Log10 transformation. The Pearson coefficient of correlation was used to analyze relationships between estradiol 17β and the diameter of the preovulatory follicle as well as among progesterone and the corpus luteum. Variation in the interval to onset of standing estrus between early-, intermediate-, and late-responding goats was analyzed using one-way ANOVA. Analyses of pooled variances among early-, intermediate-, and late-responding goats were carried out for regression of the corpus luteum, progesterone concentration, and diameter of preovulatory follicles relative to PGF2α through the generalized linear model using SPSS (version.16.0, IBM, Chicago, IL, USA).

## 3. Results

### 3.1. Response to Estrus Management

All goats (n = 19) showed standing estrus after 50.6 ± 4.8 h of PGF_2α_ treatment (100% response). The overall length (n = 19) of standup estrus was 22.3 ± 2.8 h. The overall time from the start of estrus to ovulation was 32.5 ± 4.4. The overall time to ovulation from the end of standing estrus was 9.4 ± 1.5 h (Table 1). Relative to PGF_2α_, a marked variation (*p* < 0.05) in the beginning of standing estrus and time of ovulation were detected between the goats (Table 1). The onset of standing estrus relative to PGF_2α_ was first observed in goats at 44 ± 2.0 h (early-responding; n = 7), at 51 ± 3.0 h (intermediate-responding; n = 6), and at 60 ± 0.0 h (late-responding; n = 6). The mean duration of standing estrus and the interval to ovulation from the end of standing estrus did not differ (*p* > 0.05) between the early-, intermediate-, and late-responding goats (Table 1). Similarly, the diameter of the preovulatory follicle 12 h prior to ovulation did not vary (*p* > 0.05) between the early-, intermediate-, and late-responding goats (Table 1).

### 3.2. Association between Dominant Follicle, Estradiol 17β, and Standing Estrus 

Follicular dynamics exposed that the majority of goats had two dominant follicles (13/19 = 68%) throughout the periovulatory period after PGF_2α_. The diameter of the largest dominant follicle was greater (*p* = 0.00) than second-largest dominant follicle, i.e., 7.3 ± 0.3 and 6.6 ± 0.3, respectively (Figure 1A). However, the growth rate of both the largest and second-largest dominant follicles did not differ over time until ovulation (Figure 1B). The plasma concentration of estradiol 17β increased linearly (r = 0.98, *p* = 0.003) with the mean diameter of preovulatory follicles. The highest plasma quantity of estradiol 17β was detected 12 h. previous to ovulation and standing estrus ended three hours later (Figure 2). The mean diameter of dominant follicles was larger in early-responding goats than intermediate- and late-responding goats (*p* = 0.005) following PGF_2ά_ (Figure 3). A similar pattern in the plasma concentration of estradiol 17β was observed among early-, intermediate-, and late-responding goats (Figure 4).

### 3.3. Association among Corpus Luteum, Progesterone, and Ovulation

Overall, sixty-eight percent of goats had two corpora lutea at the time of PGF_2α_ administration. The diameters of the first and second corpora lutea 12 h after PGF_2α_ were 9.6 ± 0.2 and 9.2 ± 0.6, respectively. Both corpora lutea degenerated with time (r = −0.98, *p* = 0.00) after PGF_2α_. The diameters of the corpora lutea did not vary (*p* = 0.42) from each other until ovulation (Figure 5). The plasma concentration of progesterone declined linearly (r = 0.92; *p* = 0.00) with a decline in the combined diameter of the first and second corpora lutea after PGF_2α_ (Figure 6). The regression of corpora lutea after PGF_2α_ among early-, intermediate-, and late-responding goats was different (*p* = 0.000) (Figure 7). On the other hand, plasma progesterone concentrations did not vary (*p* = 0.065) between early-, intermediate-, and late-responding goats (Figure 8).

## 4. Discussion

The current study highlights the ovarian dynamics and plasma concentration of the hormonal profile in response to single PGF_2α_ injected indiscriminately through the luteal stage in Barbari goats. Overall, one hundred percent estrus response to single PGF_2α_ was achieved in this study, which is similar to a prior report in Beetal goats synchronized with two injections of PGF2α [32]. The start of standup estrus and period to ovulation after single PGF_2α_ were also congruent with the findings in Shiba goats treated with a single dose of PGF_2α_ [19]. However, contrary to the double PGF_2α_ treatment in Beetal goats [33], both the beginning of standing estrus and interval to ovulation were longer in our study, i.e., 36.0 ± 1.2 vs. 50.6 ± 4.8 h and 66.0 ± 2.7 vs. 82.3 ± 4.0, respectively, signifying a variable reaction to single PGF_2α_ during the luteal phase in Barbari goats. On the other hand, the overall estrus response of 100%, start of standing estrus of 50.6 ± 4.8 vs. 49.2 ± 3.7 h, and interval to ovulation of 82.3 ± 4.0 vs. 77.2 ± 3.4 h in our study are comparable with a previous study in Beetal goats using single PGF_2α_, injected upon confirmation of the mature corpus luteum using ultrasonography [17].

A longer period to estrus and ovulation following PGF_2α_ was correlated with deviation in follicle expansion during the luteal stage [34]. Previous findings in goats suggest that the growth of follicles during the early or late luteal phase influences the consequences of PGF_2α_ [18,35]. In this context, goats showing early onset of estrus, in our study, had large follicles at the time of PGF_2α_ administration as compared to the late-responding goats (Figure 3). This observation is further substantiated by the evidence that the peak plasma concentration of estradiol 17β was first identified in early- rather than intermediate- or late-responding goats following the treatment (Figure 4).

During this study, PGF_2α_ was injected indiscriminately upon determining the presence of corpora lutea rather than on the exact day of the estrous cycle. As a result, goats may have been on approximately the 5th to 17th days of the estrous cycle when the treatment was initiated. According to a previous study in goats, luteolysis occurred rapidly when induced during the early (day 5) or late (day 16) luteal phase than the mid-luteal phase (day 11) [18,36]. In addition, goats had a higher concentration of progesterone in the mid-luteal phase than the early or late luteal phase of the estrous cycle [18]. Correspondingly, current findings suggest that the plasma progesterone concentration declined quickly following PGF_2α_ in early compared to intermediate- and late-responding goats (Figure 7 and Figure 8). Evidence suggests that large luteal area and higher plasma progesterone concentration at the time of PGF_2α_ treatment resulted in delayed luteolysis in goats [37]. Moreover, presence or absence of the dominant follicle at the time of PGF_2α_ administration influenced the estrus response [38]. Hence, we imply that early-responding goats were more efficient than other goats due to the smaller size of the corpus luteum, diminished progesterone concentration, and presence of large dominant follicles, as shown in Figure 3, Figure 7 and Figure 8, respectively. Moreover, the start of standing estrus to ovulation of 32.5 ± 4.4 vs. 28.6 ± 3.8 h, number of ovulations of 1.7 ± 0.4 vs. 1.95 ± 0.2, and diameter of preovulatory follicles of 7.1 ± 0.3 vs. 7.15 ± 0.3 mm in Barbari goats was also congruent with the finding in Beetal goats [17]. In short, our results consistently but indirectly support the notion that the estrus response in terms of the beginning of estrus and interval to ovulation might differ in Barbari goats due to different stages of the luteal phase at the time of PGF_2α_ treatment. However, the scope of the research can be increased if PGF_2α_ is injected at identified days of the estrous cycle with a larger number of animals as well as frequent sampling and scanning.

In our study, estradiol 17β concentration in relation to the start of standing estrus followed a similar trend to that of Shiba goats [19]. However, an earlier decline in the concentration of progesterone was observed in Shiba goats than Barbari goats, i.e., 12 vs. 24 h. This difference is due to the extensive plasma sampling in Shiba goats. In general, the duration of estrus behavior in goats varied from 20 to 58 h depending upon age, breed, season, and presence or absence of males [39]. In this context, Barbari goats in this research had a short period (22 h) of standing heat, similar to 22 h in Angora goats [40] and 22 h in Shiba goats [19] but less than the 37 h in Boer goats [41] and 30.8 ± 3.9 h in Beetal goats [17], following PGF_2α_ treatment in the breeding season.

## 5. Conclusions

In conclusion, the discrepancy of ovarian follicular dynamics and variability in hormonal profile, like estradiol 17β and progesterone, at the initiation of synchronization protocol, specifically PGF_2α_ administration, significantly affect the outcome of reproductive physiology, estrus, and ovulation in Barbari goats. That distinction is considerably controlled by the size of the developing preovulatory follicle and corpus luteum on the day of PGF_2α_ injection. In this study, the majority of goats (68%) had two dominant follicles at the time of injection, which grow significantly (*p* = 0.00) with the passage of time and became mature preovulatory follicles, but their growth rates did not diverge (*p* = 0.124) from each other until ovulation. Similarly, most of the goats 68% had two corpus lutea at the time of synchronization, which regressed significantly (*p* = 0.00) with the passage of time, but their regression rates did not fluctuate (*p* = 0.93) from each other until ovulation. Furthermore, there was a higher correlation (r = 0.98) between the combined diameter of preovulatory dominant follicles with estradiol 17β concentration after every ultrasonographic scan. Regression of the corpus luteum (CL) and a decline in progesterone level were also highly associated (r = 0.93) until ovulation. The diameter of preovulatory dominant follicles attained maximum ovulatory size faster in early- than late-responding groups (7.5 ± 0.4 mm vs. 6.5 ± 0.3 mm). Similarly, the preliminary concentration of estradiol 17β was dissimilar among the goats, which resulted in a variable surge of estradiol 17β of 48 vs. 60 vs. 72 h in goats. Contrary to that, diameter of the corpus luteum (CL) 12 h post-PGF_2α_ shot was smaller in early than intermediate- and late-responding goats (8.4 ± 0.4 mm vs. 8.9 ± 0.5 mm vs. 10 ± 0.5 mm). Similarly, there was a lower plasma concentration of progesterone at the time of treatment in some goats, which resulted in a rapid decline in progesterone level within 24 h in early groups compared to other groups (11.2 ± 0.3 vs. 19.1 ± 0.1 vs. 21.5 ± 0.5 ng/mL). Due to this, the start of estrus, peak estradiol 17β concentration, decline in progesterone concentration, and ovulation timing were considerably diverse among different goats. Moreover, this is the first ever major study profiling follicular dynamics and ovarian steroids after induced luteolysis during the preovulatory period in Barbari goats. In future, the prospects of the research might be improved if PGF_2α_ treatment is conducted on identified days of the estrus cycle with a larger sample size, frequent blood sampling, and ultrasonography to determine a more precise timing of estrus and ovulation in Barbari goats.

## Figures and Tables

**Figure 1 vetsci-10-00624-f001:**
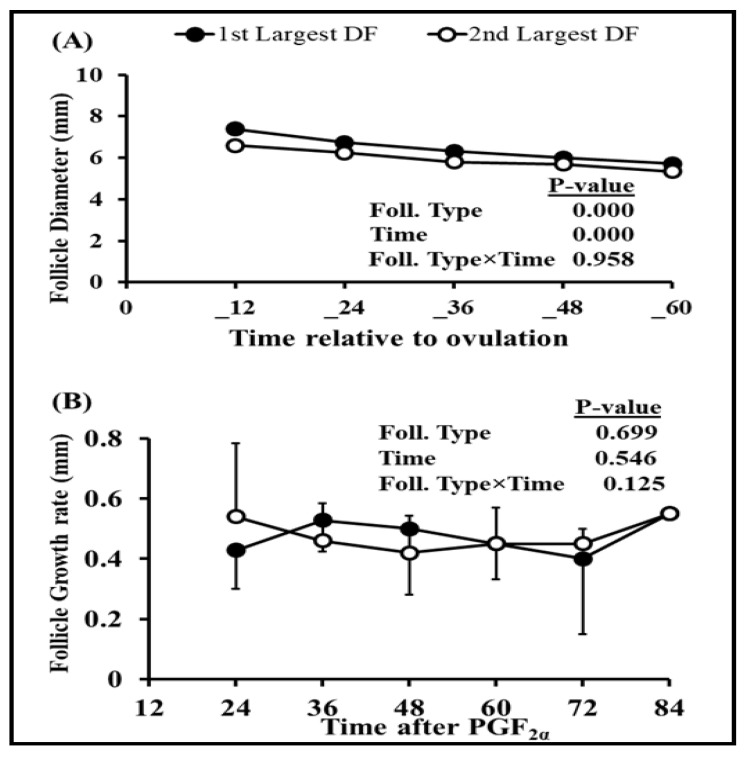
Comparison between diameters (mean ± SEM) of largest (n = 19) and second-largest dominant follicles (n = 13) relative to ovulation (0 h) indicates difference (*p* = 0.00) in diameter (**A**); conversely, the growth with time of both the largest and second-largest dominant follicles did not vary following PGF_2α_ (**B**).

**Figure 2 vetsci-10-00624-f002:**
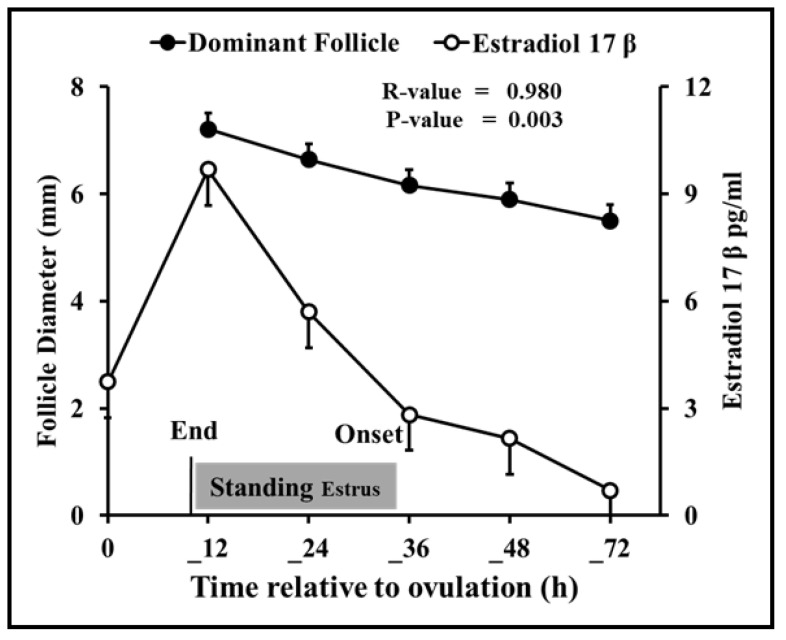
Changes in plasma level of estradiol 17β (n = 19) and size (mean ± SEM) of the dominant follicles (n = 19) were highly correlated relation to ovulation (0 h) in goats. The peak plasma level of estradiol 17β coincides with the maximum size of the dominant follicle 12 h prior to ovulation. Barbari goats (n = 19) were first observed in standing estrus 19.3 h prior to the peak plasma concentration of estradiol 17β.

**Figure 3 vetsci-10-00624-f003:**
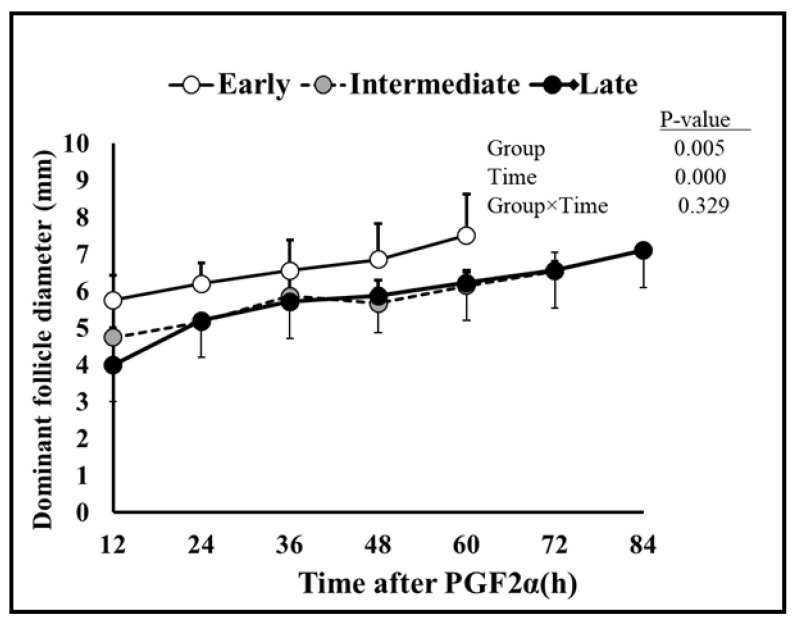
Comparison of the dominant follicle (mean ± SEM) amongst early- (n = 7), intermediate- (n = 6), and late-responding (n = 6) goats following PGF_2α_. Mean diameter of the dominant follicle was larger in early-responding than intermediate- and late-responding goats.

**Figure 4 vetsci-10-00624-f004:**
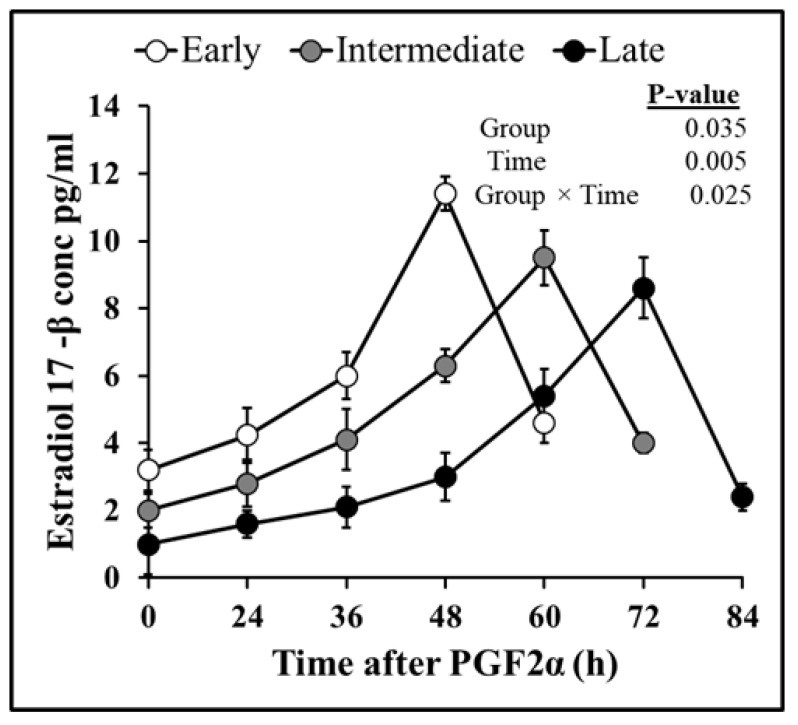
Comparison of peak plasma concentration of estradiol 17β (mean ± SEM) between early- (n = 7), intermediate- (n = 6), and late-responding (n = 6) goats following PGF_2α._ The plasma concentration of estradiol 17β peaked earlier in early-responding than intermediate- and late-responding goats.

**Figure 5 vetsci-10-00624-f005:**
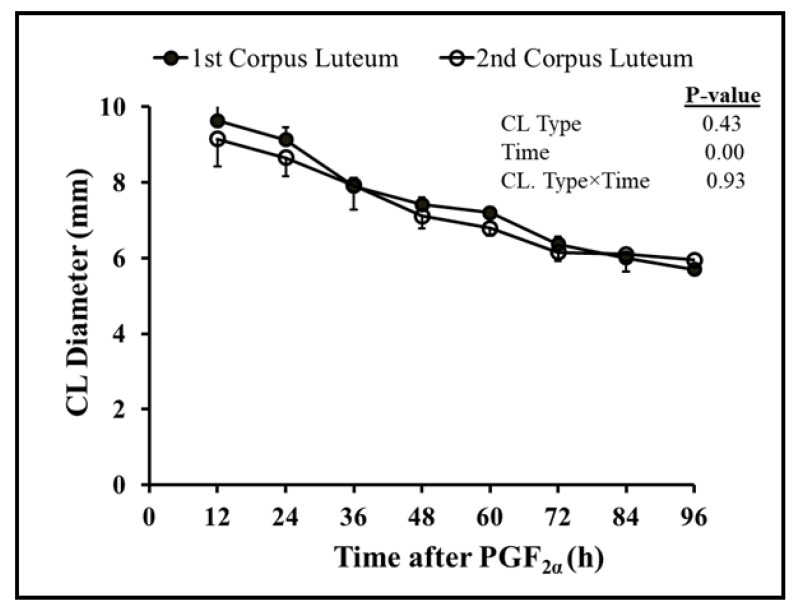
Comparison of diameters (mean ± SEM) of first (n = 19) and second (n = 13) corpora lutea indicate that the corpora lutea did not differ (*p* = 0.43) from each other after PGF_2α_ (0 h) in Barbari goats.

**Figure 6 vetsci-10-00624-f006:**
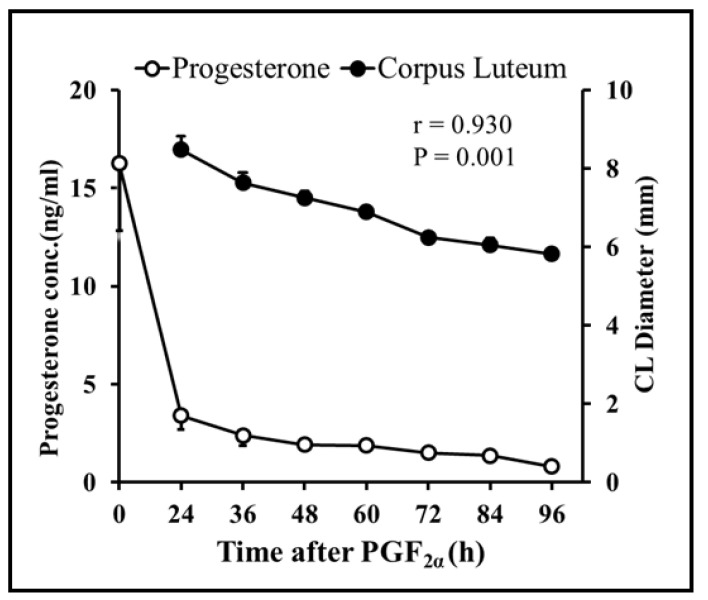
Changes in the plasma concentration of progesterone (n = 19) and mean diameter of both corpora lutea (combined) (n = 19) after PGF_2α_ were highly correlated in Barbari goats. The plasma concentration of progesterone decreased within 24 h after PGF_2α_.

**Figure 7 vetsci-10-00624-f007:**
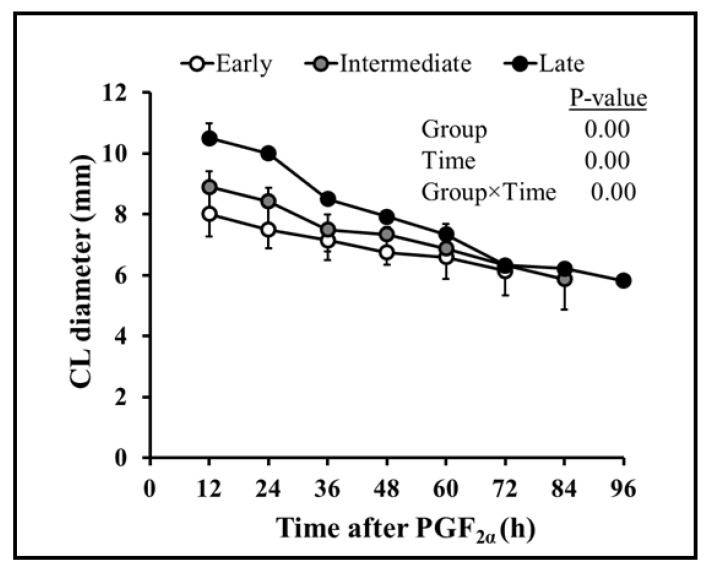
Comparison of diameter of corpora lutea (CL; mean ± SEM) among early- (n = 7), intermediate- (n = 6), and late-responding (n = 6) goats relative to PGF_2α_ (0 h). CL with the largest diameter in late-responding goats regressed slowly compared to intermediate- and early-responding goats.

**Figure 8 vetsci-10-00624-f008:**
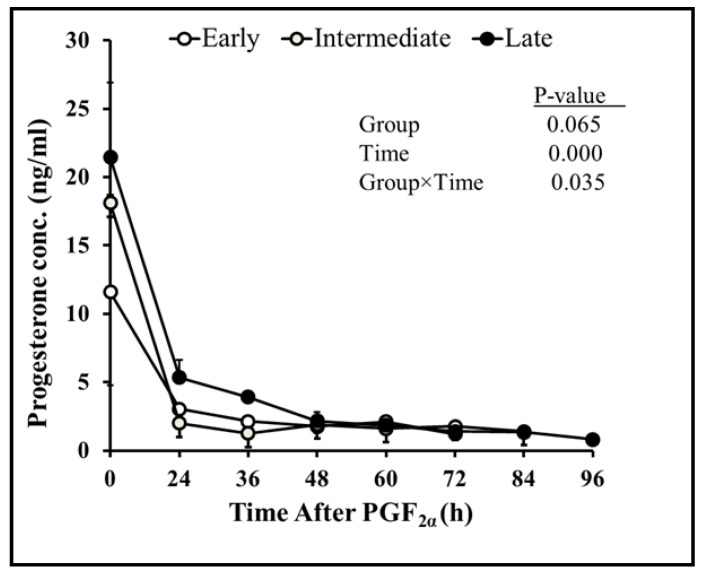
Comparison of plasma progesterone concentration (mean ± SEM) between early- (n = 7), intermediate-(n = 6), and late-responding (n = 6) goats relative to PGF_2α_ (0 h). The plasma concentration of progesterone declined within 24 h after PGF_2α_. Although changes in progesterone level did not significantly differ among groups (*p* = 0.065), changes in the plasma concentration of progesterone over time were different among early-, intermediate-, and late-responding goats (*p* = 0.03).

**Table 1 vetsci-10-00624-t001:** Response to estrus synchronization following a single injection of prostaglandin F_2α_ (PGF_2α_) administered indiscriminately through the luteal stage of the estrus cycle in Barbari goats (n = 19).

	Relative Responses after PGF_2α_			
	Overall (n = 19)	Early (n = 7)	Intermediate (n = 6)	Late (n = 6)
Onset of standing estrus (h)	50.6 ± 4.8	44 ± 2.0 ^a^	51 ± 3.0 ^b^	60 ± 0.0 ^c^
Interval to ovulation (h)	82.3 ± 4.0	72 ± 0 ^a^	84 ± 0 ^b^	96 ± 0 ^c^
Start of estrus to ovulation Diameter of corpus luteum 12 h post-PGF_2α_ (mm)	32.5 ± 4.4 8.9 ± 0.4	28 ± 1 8.4 ± 0.4	33 ± 1.5 8.9 ± 0.5	36 ± 0 10 ± 0.5
Diameter of preovulatory follicle (mm) 12 h prior to ovulation	7.1 ± 0.3	7.5 ± 0.4	6.5 ± 0.3	7.1 ± 0.1
Duration of standing estrus (h)	22.3 ± 2.8	18 ± 3.5	24 ± 0.0	27 ± 3.0
Interval to ovulation from end of standing estrus (h)	9.4 ± 1.5	10 ± 0.2	9 ± 3.0	9 ± 3.0

^a, b, c^ denote difference at *p* < 0.05 among groups.

## Data Availability

All datasets collected and analyzed during the current study are available from the corresponding author upon reasonable request.

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
