# Peer review of "Ovarian Dynamics and Changes in Estradiol-17β and Progesterone Relationship with Standing Estrus, Preovulatory Follicles, and Ovulation Using Single Prostaglandin F2α Injection in Barbari Goats"

_vetsci, 2023, doi:10.3390/vetsci10100624_

Round 1

Reviewer 1 Report

Do you know the age of the goats? Is the number of lambings prior to the experience of the goats known?

You should improve the presentation of the graphs, which right now are low quality images.

Repetitions are needed to have more reliable conclusions.

Author Response

Response to Reviewer .

Please see the attachment below.

Reviewer 2 Report

Does the introduction provide sufficient background and include all relevant references?

Ref. 8 – report, convert to publication if possible

Ref. 9 - the publication refers to cattle, the text suggests that the cited publication refers to goats. Please add that the publication concerns cattle or change the reference.

Ref. 14 - applies only to goats, does not apply to cattle - complete the citation with cattle.

Is the research design appropriate?

The authors describe a narrow aspect of reproductive physiology or biotechnology in one breed of goat. The research was performed on a small population of 19 animals, additionally divided into 3 groups. Does such a low number justify the description of the entire breed? What methods were used to make the results on small groups reliable for the reader?

Are the conclusions supported by the results?

The conclusions presented in the summary and conclusions section are different and not related to each other. The conclusions contained in the conclusions section are only partially supported by the results and are poorly related to the content of the article. The conclusions are not a summary of research and results, but the authors' guesses and assumptions. They definitely need a major change!!!

The text is well written and understandable. Did the authors perform language proofreading with a certificate? In my opinion, the text requires minor linguistic corrections, please have it proofread, e.g. by a native speaker.

Author Response

Response to Reviewer 2.

Please see the attachment below.
